# Circular and Fusion RNAs in Medulloblastoma Development

**DOI:** 10.3390/cancers14133134

**Published:** 2022-06-26

**Authors:** Ani Azatyan, Peter G. Zaphiropoulos

**Affiliations:** Department of Biosciences and Nutrition, Karolinska Institutet, 14183 Huddinge, Sweden; ani.azatyan@ki.se

**Keywords:** brain cancer, pediatric cancer, back-splicing, read-through transcription, trans-splicing

## Abstract

**Simple Summary:**

Expression of circular RNAs is known to be deregulated in cancer. Here the most comprehensive set of differentially expressed RNA circles in medulloblastoma compared to cerebellum is provided. Additionally, fusion RNAs are also identified in both cancerous and normal cerebellar tissue. Some of the fusions detected in medulloblastoma are generated by genomic rearrangements that link different genes. However, fusion RNAs are also detected in normal cerebellum. In fact, there are cases where the same fusion RNA is also found in medulloblastoma. This observation underscores that the formation of fusion transcripts may not be limited to chromosomal events but could also result from mechanisms that act at the RNA level. These include read-through transcription of neighboring genes and intermolecular splicing of pre-mRNAs from different genes Importantly, these RNA “recombination” events occur not only in normal but also in cancerous tissue.

**Abstract:**

*Background.* The cerebellar cancer medulloblastoma is the most common childhood cancer in the brain. *Methods*. RNA sequencing of 81 human biospecimens of medulloblastoma using pipelines to detect circular and fusion RNAs. Validation via PCR and Sanger sequencing. *Results.* 27, 56, 28 and 11 RNA circles were found to be uniquely up-regulated, while 149, 7, 20 and 15 uniquely down-regulated in the SHH, WNT, Group 3, and Group 4 medulloblastoma subtypes, respectively. Moreover, linear and circular fusion RNAs containing exons from distinct genes joined at canonical splice sites were also identified. These were generally expressed less than the circular RNAs, however the expression of both the linear and the circular fusions was comparable. Importantly, the expression of the fusions in medulloblastoma was also comparable to that of cerebellum. *Conclusion**s.* A significant number of fusions in tumor may be generated by mechanisms similar to the ones generating fusions in normal tissue. Some fusions could be rationalized by read-through transcription of two neighboring genes. However, for other fusions, e.g., a linear fusion with an exon from a downstream gene joined 5′ to 3′ with an exon from an upstream gene, more complicated splicing mechanisms, e.g., trans-splicing, have to be postulated.

## 1. Introduction

Medulloblastomas are malignant cerebellar tumors occurring in infants, children and young adults. Proper diagnosis encompasses magnetic resonance imaging, with the treatment options including resection, chemotherapy, and craniospinal radiation [1,2,3]. Despite recent advances in therapeutic outcomes the survival rate of medulloblastoma patients has reached plateau levels [4]. Approximately one third of patients eventually die from this disease, while survivors may suffer from long-term side effects from the implemented therapeutic approaches. It is thought that an improved understanding of medulloblastoma biology may allow the development of novel treatment strategies. In this direction, gene expression profiling has categorized medulloblastomas into 4 subgroups, the SHH (Sonic Hedgehog), WNT (Wingless), Group 3, and Group 4 [5].

Exons are usually joined in a 5′ to 3′ direction giving rise to mature linear RNA molecules. However, in recent years the possibility that a 3′ exon is back-spliced to a 5′ exon has gained attention. This generally gives rise to a circular RNA containing these exons, and is particularly stable, since it cannot be attacked by exonucleases [6,7,8,9,10]. There is an increasing number of cases where circular RNAs have been claimed to exert numerous biological functions ranging from miRNA sequestration to protein synthesis via internal ribosomal entry sites [11,12,13,14,15]. In fact, there is convincing evidence that circular RNA expression is de-regulated in cancer [16]. Moreover, in colorectal, hepatic, prostate and bladder cancer, de-regulated circular RNAs have been demonstrated to act as oncogenes or tumor suppressor genes [17,18,19,20]. However, there is little information on circular RNA expression in medulloblastoma [21,22].

There is also evidence that linear RNA molecules can be produced where a 3′ exon is found spliced upstream of a 5′ exon. The original observations are over twenty years old [23,24], but have also been identified via next generation sequencing, albeit at a lower frequency than RNA circles [22,25]. The underlying mechanism is considered to be a trans-splicing event where a 3′ exon from one transcript is joined to a 5′ exon from another transcript from the same gene.

There are also data supporting rare events that splice together transcripts from different genes in the same genomic locus [26,27,28]. Moreover, in normal tissues splicing of gene transcripts from different chromosomes has been observed and related with genomic recombination in cancer [29,30]. Additionally, high-throughput sequencing has expanded the number of fusion transcripts in normal tissues, some of which are also found in cancer [31,32].

Here we analyzed, via RNA-seq 81, medulloblastoma samples encompassing all 4 subtypes, as well as 5 samples of normal cerebellum. Implementation of RNA isolation protocols that do or do not enrich for circular RNAs and of various bioinformatic pipelines resulted in the identification of not only RNA circles encompassing exons from single genes but also of linear and circular RNAs encompassing exons from different genes.

## 2. Materials and Methods

### 2.1. RNA Sample Information

23, 6, 17 and 35 RNA samples from the SHH, WNT, Group 3 and Group 4 medulloblastoma subgroups, respectively, were provided by the Children’s Brain Tumor Network (https://cbtn.org) (accessed on 1 March 2022), and 5 normal cerebellum RNA samples were purchased from Takara (Kusatsu, Japan), Amsbio (Abingdon, UK) and BioChain (Newark, CA, USA) (Appendix A).

### 2.2. RNA-Seq Library Preparations

The RNA samples were subjected to poly(A) and circular RNA enriched library preparations as described previously [22]. The concentration and quality of the cDNA libraries were assessed with Qubit (Life Technologies, Carlsbad, CA, USA) and TapeStation (Agilent Technologies, Santa Clara, CA, USA), respectively. cDNA library preparations from 17 circular RNA enriched samples did not meet quality control criteria and were omitted from further analyses (Appendix A). The cDNA libraries were sequenced on the Illumina NovaSeq 6000 S4 platform (150 cycles; paired-end reads; ≈25 M reads per sample).

### 2.3. RNA-Seq Data Analysis

mRNA detection: reads from the poly(A) enriched preparations were aligned to the Gencode GRCh38 reference genome assembly using the STAR (v2.5.2) aligner. Counts were assigned to genes using featureCounts (v1.5.1).

Circular RNA detection: reads from the circular RNA enriched preparations were analyzed with the CIRCexplorer2 to identify back-spliced junctions and aligned to Gencode GRCh38 using CIRCexplorer2/STAR [33] with the default parameters.

mRNA and circular RNA counts (raw reads) were analyzed with the DESeq2 Bioconductor package (v1.32.0) using the apeglm (v1.14.0) method for effect size shrinkage and log fold change estimations [34] in R software (R-v4.1.1). As the circular RNA datasets had an abundance of low reads, RNA circles with raw mean counts lower than 2 were filtered out. Gene expression level comparisons (as log2 fold change estimation of DESeq2 normalized counts) were performed using the Wald test statistics as implemented in DESeq2.

Fusion detection: reads from the poly(A) enriched and circular RNA enriched preparations were aligned to Gencode GRCh38 using STAR-Fusion [35] with the default parameters.

Due to the very low expression of the fusion reads, it was impossible to calculate size factors and effectively perform gene expression level comparisons with the DESeq2 method as done with the circular RNA and mRNA datasets. Consequently, we simply pinpointed the fusion transcripts with the highest reads.

### 2.4. Validation of Fusion Transcripts by Sanger Sequencing

The NCOA2--TRAM1 and FBXO25--SEPTIN14 fusion transcripts were validated by PCR using oligo(dT) primers as described previously [22]. Primers to detect the NCOA2--TRAM1 fusion junction were: Forward-5′-GCCTCGGCTACAGCTTCGG, Reverse-5′-AGGTTATGGGGATAAGCCCTCCA, and the FBXO25--SEPTIN14 fusion junction: Forward-5′-AGGCATGGCTATTGCACCTTG, Reverse-5′-CGAGCTTCCTTATCTCCTCCTGT.

## 3. Results

### 3.1. Circular RNAs

81 medulloblastoma and 5 cerebellar RNA samples (Appendix A) were first subjected to standard RNA-seq (Appendix A). 65 of the medulloblastoma and 4 of the cerebellar RNA samples could effectively be enriched for circular RNAs via ribosomal RNA depletion followed by RNAse R treatment (Section 2 Materials and Methods). The CIRCexplorer2 pipeline [33] was implemented on the RNA-seq data of the circular RNA enriched preparations to identify back-spliced junctions. Using the stringent cutoffs of padj < 0.05 and |log2FoldChange| > 1, circular RNAs differentially expressed in the SHH, WNT, Group 3 and Group 4 medulloblastoma subtypes compared to cerebellum were identified (Figure 1, Appendix A). Worth noting is that in the SHH subtype the uniquely down-regulated circles vastly exceeded the up-regulated, 149 vs. 27, while the opposite was true for the WNT subtype, 7 vs. 56 (Figure 1B). In general, the expression of circular RNAs was found to be substantially lower than the expression of their corresponding linear mRNAs, apart from the cases of the RMST, RIMS1 and CARM1P1 genes where it was more comparable (Appendix A).

Moreover, circular RNAs differentially expressed in each medulloblastoma subtype compared to the other three subtypes (including cerebellum) were also identified (Appendix A). These lists included a substantial number of the uniquely de-regulated circles of Figure 1B, detailed in Appendix A. providing statistical significance in that context.

Samples 2, 5, 6 and 9 of the SHH medulloblastoma subtype are progressive tumors, with the initial tumors being samples 3, 4, 7 and 10, respectively (Appendix A). Using the same stringent cutoffs of padj < 0.05 and |log2FoldChange| > 1, a predominance of down-regulated to up-regulated mRNAs, 90 vs. 32, in the progressive compared to the initial tumors was observed (Appendix A). However, no de-regulated circular RNAs were identified, possibly because progressive sample 9 was not effectively enriched in circular RNAs and consequently excluded from further analysis, along with the corresponding initial sample 10, reducing the samples to three pairs. Implementation of the more relaxed cutoff of *p* < 0.05 and |log2FoldChange| > 1 identified 1 down-regulated and 12 up-regulated circular RNAs (Appendix A). Of note, from these 122 and 13 de-regulated mRNAs and circular RNAs only 33 and 1, respectively, were present within the de-regulated mRNAs and circular RNAs in the SHH medulloblastoma subtype compared to cerebellum (Appendix A), with some, e.g., circular RNA AKT3, being de-regulated in the opposite direction.

### 3.2. Fusion RNAs–Cerebellum

To identify fusion transcripts the STAR-Fusion pipeline [35] was implemented on the data obtained from the circular and the poly(A) enriched RNA preparations in cerebellum and medulloblastoma. Overall, the number of fusion junctions (Appendix A) was lower than that of the back-spliced junctions (Appendix A), arguing for a predominance of circular relative to fusion RNAs. Importantly, the raw counts of the fusions from the circular enriched preparations were found to be comparable to those of the fusions from the poly(A) enriched preparations (Figure 2A, Appendix A). This is in contrast to the general observations of circular RNAs being expressed less than linear poly(A) mRNAs [36]. The 20 most abundant fusion transcripts from each of the linear and circular RNA preparations in cerebellum and medulloblastoma were selected (Appendix A). These were further filtered for validated splice sites at fusion junctions in validated exons of NCBI transcripts within the curated RefSeq database (Appendix A). This resulted in a list of 5 linear and 1 circular fusions in cerebellum and 4 linear and 3 circular fusions in medulloblastoma (Table 1, Figure 2B,C).

The two most abundant of the 5 selected linear fusions in cerebellum that originate from protein coding genes, namely FBXO25--SEPTIN14 and NCOA2--TRAM1 (Table 1, Appendix A), were validated by PCR and Sanger sequencing (Figure 3, Appendix A). Interestingly, the SEPTIN14 sequence in the fusion has an A nucleotide instead of G at position 12. This suggests that the origin of that sequence may not be the SEPTIN14 gene at chromosome 7 but one of the expressed SEPTIN14 pseudogenes P6 (NR_109817.1) or P20 (NR_040415.1) in chromosomes 6 and 20, respectively, both of which are identical to SEPTIN14 in the sequenced region except for that A nucleotide. Irrespective of what may be the true origin, the fact remains that transcripts from the FBXO25 gene in chromosome 8 are spliced to transcripts from a different chromosome.

Importantly, the NCOA2 gene is downstream of TRAM1 in chromosome 8 (Table 1, Appendix A). Consequently, the validated NCOA2--TRAM1 linear fusion can not originate from read-through transcription beyond the end of the TRAM1 gene that extends into the NCOA2 gene. Rather more complicated splicing mechanisms, e.g., trans-splicing, have to be involved, in line with the mechanisms required to rationalize fusions from exons in different chromosomes.

Two of the 5 selected linear fusions in cerebellum, namely SEPTIN7P14--PSPH and PAUPAR--RCN1, were also found in medulloblastoma. However, none of these 5 selected fusions were found in the circular enriched RNA preparations (Figure 2B, Appendix A).

The selected circular fusion ADAMTSL3--SH3GL3 in cerebellum originates from two neighboring genes in the same orientation at chromosome 15 (Table 1, Appendix A). Conceivably, read-through transcription from the SH3GL3 gene extends into the ADAMTSL3 gene, followed by back-splicing that joins the end of ADAMTSL3 exon 3 to the start of SH3GL3 exon 7 (Figure 4A). Supporting the claim of SH3GL3 transcripts extending into the ADAMTSL3 gene is the fact that two SH3GL3--ADAMTSL3 linear fusions have recently been identified in normal testis [32]. Worth noting is that the ADAMTSL3--SH3GL3 circular fusion is also found in medulloblastoma (Figure 2B), along with two other ADAMTSL3--SH3GL3 circular fusions encompassing the same SH3GL3 exon and one SH3GL3--ADAMTSL3 circular fusion encompassing the same ADAMTSL3 exon (Appendix A). However, the ADAMTSL3--SH3GL3 fusion was not found in the linear enriched RNA preparations (Appendix A).

### 3.3. Fusion RNAs–Medulloblastoma

The most abundant of the selected linear fusions in medulloblastoma is the KANSL1--ARL17A (Table 1, Appendix A). This has been observed in acute lymphoblastic leukemia [37,38] but also recurrently in normal tissues [31]. The KANSL1 gene is downstream to the ARL17A gene in chromosome 17, which prohibits a transcriptional read-through mechanism (Appendix A). Importantly, the same exons of this linear fusion are also found in the most abundant selected circular fusions in medulloblastoma, namely the ARL17A--KANSL1 and the KANSL1--ARL17A (Table 1, Appendix A). Since no linear ARL17A--KANSL1 fusions are observed (Appendix A), the circular fusions with exon 3 of ARL17A spliced to exon 3 of KANSL1 and with exon 3 of KANSL1 spliced to exon 3 of ARL17A may represent the same circle, which could be composed solely of these two exons. This is supported by the fact that medulloblastoma samples expressing the ARL17A--KANSL1 circular fusion generally also express the KANSL1--ARL17A circular fusion, in fact, 11 out of 13 cases (Figure 2C, Appendix A). Moreover, out of these 11 medulloblastoma samples, 9 also express the linear KANSL1--ARL17A fusion (Appendix A), suggesting that the mechanisms generating the KANSL1--ARL17A linear and circular fusions may be related.

Additional selected linear fusions in medulloblastoma that cannot be interpreted via read-through transcription are the TFG--ADGRG7 and the PVT1--CASC8 (Table 1, Appendix A). The former is similar to the cases described above, with an exon from a downstream gene spliced to an exon from an upstream gene. Importantly, the same TFG--ADGRG7 fusion is found in the analysis of 53 different normal tissues [32]. Moreover, the same TFG--ADGRG7 fusion is also found in the circular RNA preparations (Figure 2C, Appendix A). In fact, the 2 medulloblastoma samples expressing the TFG--ADGRG7 circular fusion also express the TFG--ADGRG7 linear fusion (Appendix A). This supports the claim that exons from a downstream gene can splice to exons from an upstream gene, with the process being able to generate not only linear but also circular fusions, as seen with KANSL1--ARL17A.

However, in the latter case, PVT1 and CASC8 are in opposite strands and in a head to head orientation (Table 1, Appendix A). Interestingly, this fusion is present in only two samples, with one (sample 67) having the highest number of all fusion reads, 2620, out of which 1811 are the selected PVT1--CASC8 fusion, with other reads in the sample having the same PVT1 exon spliced to additional exons in the same locus (Appendix A). Moreover, in another medulloblastoma sample (sample 74), the same PVT1 exon is spliced to a number of exons about 35Mb away from the PVT1 locus (Appendix A). Importantly, PVT1, a non-coding oncogene, neighbors the MYC oncogene (Appendix A), with both genes known to be amplified in cancer [39], and the medulloblastoma samples harboring the observed PVT1 fusions belong to the Group 3 medulloblastoma subtype, characterized by MYC amplifications [40].

For the last selected circular fusion in medulloblastoma, the ARL17B--KANSL1, it should be noted that the coding sequences of ARL17A (NM_001113738.2) and ARL17B (NM_001039083.5), including the exon 3 at the fusion, are identical. Consequently, the pipeline cannot, in principle, discriminate which of these tandem and in the same orientation genes is spliced to the KANSL1 gene (Appendix A). Moreover, this observation increases the number of medulloblastoma samples expressing both the ARL17A/B--KANSL1 and the KANSL1--ARL17A/B circular fusions from 11 to 14, with all of the 3 new samples expressing the linear KANSL1--ARL17A/B fusion (Appendix A), further supporting the interrelation of linear and circular fusions in this context.

Worth noting is that in the three documented linear fusions having exons from a downstream gene spliced to exons from an upstream gene, namely, NCOA2--TRAM1, KANSL1--ARL17A/B, and TFG--ADGRG7, the expression of the downstream gene is higher than the upstream (Appendix A). This is particularly pronounced in the last two cases, with KANSL1 and TFG having at least an order of magnitude higher expression than ARL17A/B and ADGRG7, respectively (Appendix A).

## 4. Discussion

Here the profile of circular and fusion RNA expression in medulloblastoma was interrogated. Using stringent statistical criteria, 367, 185, 188 and 210 circular RNAs de-regulated in the SHH, WNT, Group 3 and Group 4 subtypes of medulloblastoma, respectively, in comparison to cerebellum, were identified. Earlier studies have detected 3 up- and 30 down-regulated circular RNAs in medulloblastoma [21] and 17 circular RNAs down-regulated in the SHH subtype of medulloblastoma [22]. Worth noting is that an abundant circular RNA originating from the RMST gene was found to be down-regulated in the SHH but up-regulated in the WNT subtype (Figure 1C,D, Appendix A). Additionally, there were 579, 135, 69 and 347 de-regulated RNA circles in the SHH, WNT, Group 3 and Group 4 subtypes, respectively, in comparison to both the remaining subtypes and to cerebellum (Appendix A). The uniquely de-regulated circles (Figure 1B) found in these lists (Appendix A) have a good potential as possible biomarkers for subtype-specific disease development [36].

Moreover, using stringent cutoffs, linear and circular fusion RNAs were identified in both cerebellum and medulloblastoma. The presence of fusion RNAs in normal tissue, some of which were also validated by PCR and Sanger sequencing, raises questions as to whether chromosomal rearrangements are involved. In general, DNA in normal cells is considered stable. However, it is still conceivable that, in a small number of cells present in the sample analyzed, such an event might have occurred. This would be in line with the low number of reads for the fusion RNAs in comparison to the much higher number of reads, observed in standard RNA-seq, for the mRNAs of the two genes that are involved in the fusion.

On the other hand, there are alternative scenarios that are also compatible with the observed low number of fusion reads. In cases where the genes are in the same locus and in the same orientation some transcripts from the first gene may extend into the second gene. Forward splicing of these read-through transcripts can generate fusions with exons of the first gene spliced to exons of the second gene, as seen in the linear PAUPAR--RCN1 and RPS10--NUDT3 fusions. Moreover, back-splicing of the read-throughs can generate circular RNAs with exons from the second gene spliced to exons from the first gene, as seen in the circular ADAMTSL3--SH3GL3 fusion.

However, linear fusions have also been observed from genes in the same locus and in the same orientation, but with exons from the second gene spliced to exons from the first gene, e.g., NCOA2--TRAM1. In these cases, a read-through mechanism can not interpret the observations. Consequently, scenarios that involve intermolecular splicing of transcripts from different genes, i.e., trans-splicing, have to be postulated. Such mechanisms may also rationalize splicing from genes in different chromosomes/loci or in opposite orientations from the same locus [28,29,30].

In tumor tissues, it is well-known that chromosomal rearrangements can drive tumor growth [41,42]. Therefore, the fusions observed in medulloblastoma may result from such events. On the other hand, it should be noted that the overall number of fusion reads in the medulloblastoma compared to the cerebellar samples is quite similar, apart for a few outliers (Figure 2A). The outlier with the highest number of fusion reads in medulloblastoma is sample 67 from the Group 3 subtype, with most of these reads originating from the PVT1--CASC8 linear fusion (Appendix A). The PVT1 gene, positioned in proximity to the MYC oncogene (Appendix A), is known to engage in chromosomal events and drive tumorigenesis [43]. On the other hand, the outlier with the second highest number of fusion reads in medulloblastoma is sample 37 from the Group 4 subtype, with most of these reads originating from a KHDRBS2-OT1--KHDRBS2 circular fusion (Appendix A). Close scrutiny, however, reveals that the “KHDRBS2-OT1 gene” is simply an alternative transcript of the annotated RefSeq gene KHDRBS2 (www.ncbi.nlm.nih.gov/gene/202559) (accessed on 1 March 2022). In fact, a KHDRBS2 circular RNA, highly expressed in Group 4 medulloblastomas, has been identified in our analysis (Appendix A). Thus, some but not all outliers in the medulloblastoma fusion reads may indeed represent chromosomal events. However, the majority of the medulloblastoma samples have fusion read counts that are comparable to cerebellum. Moreover, selected fusions found in cerebellum were also present in medulloblastoma, i.e., SEPTIN7P14--PSPH, PAUPAR--RCN1 and ADAMTSL3--SH3GL3, and selected fusions in medulloblastoma are known to be expressed in normal tissues, i.e., KANSL1--ARL17A/B and TFG--ADGRG7, leaving open the possibility that molecular events that can generate fusion transcripts in normal tissue, e.g., read-through transcription or trans-splicing, may also act in tumors.

Of particular interest is the KANSL1--ARL17A/B fusion, as both linear and circular forms are observed in numerous medulloblastoma samples. In fact, the KANSL1 exon 3 of the fusion is also involved in back-splicing events generating two KANSL1 circular RNAs, the only two KANSL1 circles identified in our analysis (Appendix A). Exon 3 is preceded by an over 76 kb intron, the largest of the KANSL1 gene (Appendix A), and may be especially prone to non-typical splicing events. These may allow complex trans-splicing/back-splicing interactions resulting in not only linear but also circular fusions (Figure 4B). The fact that KANSL1 is expressed to a much higher extent than ARL17A/B may have a role in the directionality of the fusion, i.e., KANSL1--ARL17A/B but not ARL17A/B--KANSL1 linear fusions are detected. A similar scenario is seen with the presence of a TFG--ADGRG7 but not a ADGRG7--TFG linear fusion and the much higher expression of TFG relative to ADGRG7.

## 5. Limitations

The number of RNA samples in the cerebellar and the WNT medulloblastoma groups is small, at 5 and 6, respectively, and this can impact the robustness of the RNA-seq analysis. Additionally, the abundance and the number of fusion RNAs detected is quite small and this does not permit an effective statistical analysis of differential expression in cerebellum versus the medulloblastoma subtypes. Increasing the read depth, currently at 25 M reads per sample, should provide the means to overcome this limitation. It is also possible that certain fusion RNAs, similarly to certain circular RNAs, may have limited impact on disease development and could simply represent “passenger” molecules, reflecting the stochasticity of the gene expression and splicing processes [22,36]. Even in such a scenario, differentially expressed circles, due to their stability, could still be effective as diagnostic markers [36,44].

## 6. Conclusions

Here, de-regulated circular RNAs in the four subtypes of medulloblastoma have been identified. Additionally, linear and circular fusions RNAs have also been observed. For some of these, e.g., PVT1--CASC8, chromosomal events are implicated. However, for a large number of these fusions, a plausible interpretation is molecular mechanisms similar to the ones generating fusions in normal tissue, i.e., read-through transcription and trans-splicing.

## Figures and Tables

**Figure 1 cancers-14-03134-f001:**
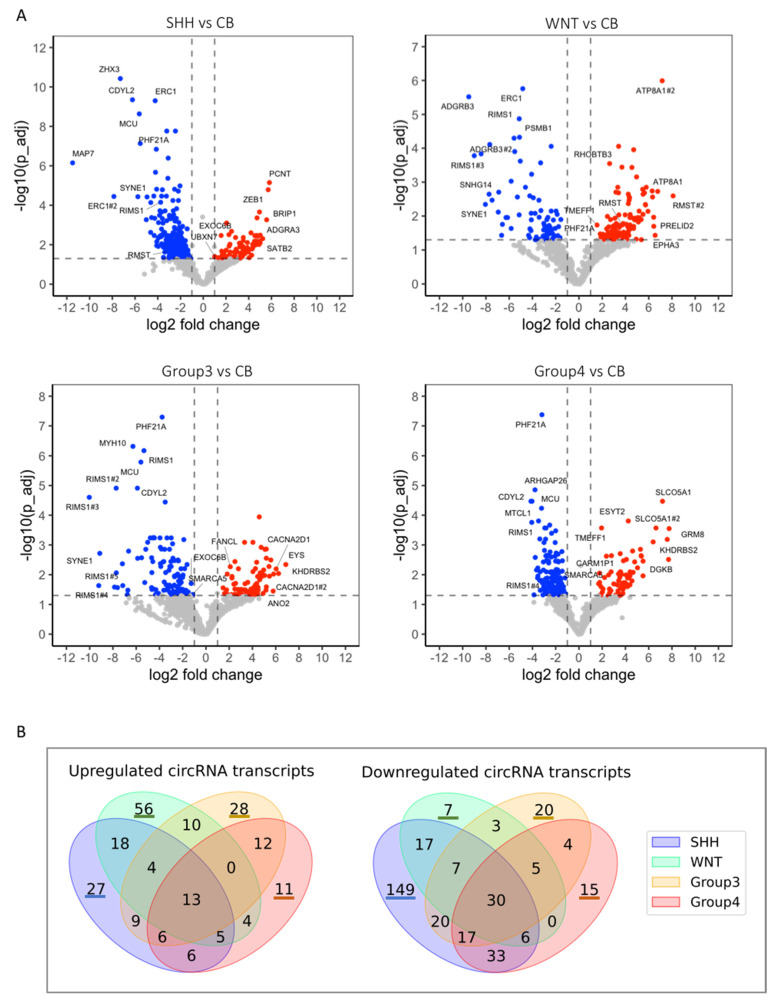
Differentially expressed circular RNAs in human medulloblastoma and normal cerebellum. (**A**) Volcano plots of the RNA-seq data of the SHH, WNT, Group 3 and Group 4 medulloblastoma tumor and normal cerebellum samples. Cutoff thresholds are assessed with the DESeq2 method, where the Wald significance test is applied as |log2 fold change| > 1, padj < 0.05. Annotated are selected circular RNAs defined as the top 5 with highest normalized mean count across samples, the top 5 with lowest padj value, the top 5 with highest log2 fold change and the top 5 with lowest log2 fold change. (**B**) Venn diagrams depicting uniquely and commonly up- and down-regulated circular RNAs (circRNA) in SHH, WNT, Group 3 and Group 4 medulloblastoma subtypes compared to normal cerebellum. (**C**–**E**) Violin plots of the expression of uniquely up- or down-regulated circular RNAs (circRNA) in the SHH (**C**), WNT (**D**), Group 3 (**E**) and Group 4 (**F**) medulloblastomas that are also annotated as selected circular RNAs in (**A**).

**Figure 2 cancers-14-03134-f002:**
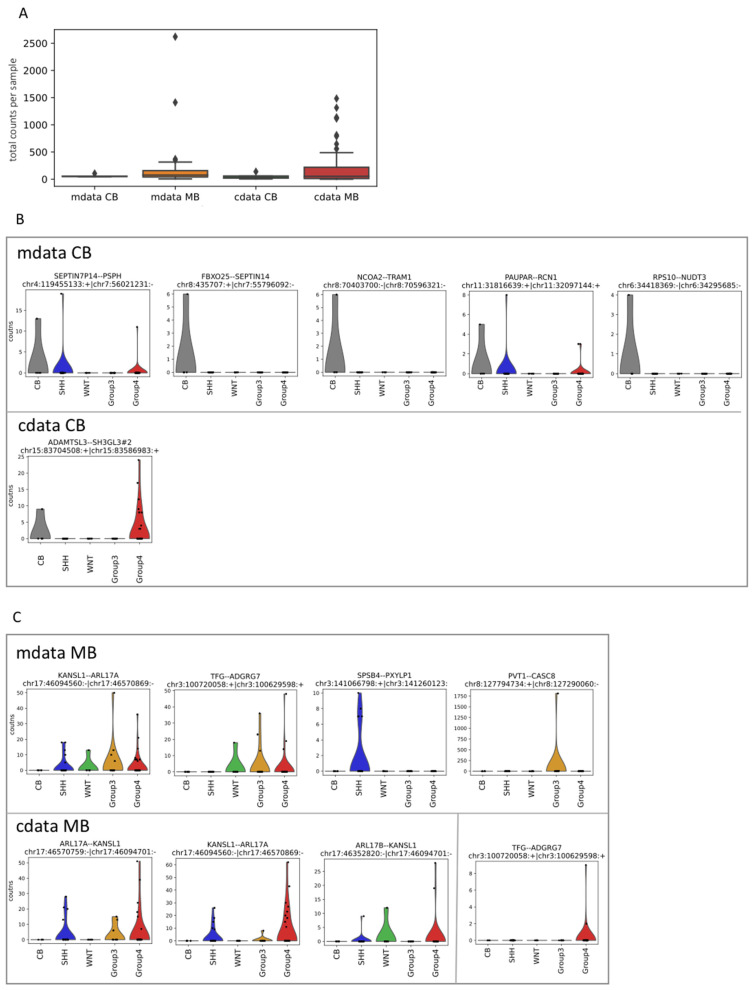
Fusion transcripts in linear and circular RNAs from cerebellum and medulloblastoma. (**A**) Box plots of the expression of linear (mdata) and circular (cdata) fusion transcripts in cerebellum (CB) and medulloblastoma (MB). The y-axis shows the total counts per sample of all fusion transcripts (black rhombi indicate outlier samples). (**B**,**C**) Violin plots of the expression of the selected fusions (Table 1) in the linear (mdata) and circular (cdata) datasets from cerebellum (**B**) and medulloblastoma (**C**). Note that the selected TFG--ADGRG7 linear fusion in medulloblastoma is also found in the circular dataset. The lack of expression of a selected fusion in a dataset is not shown.

**Figure 3 cancers-14-03134-f003:**
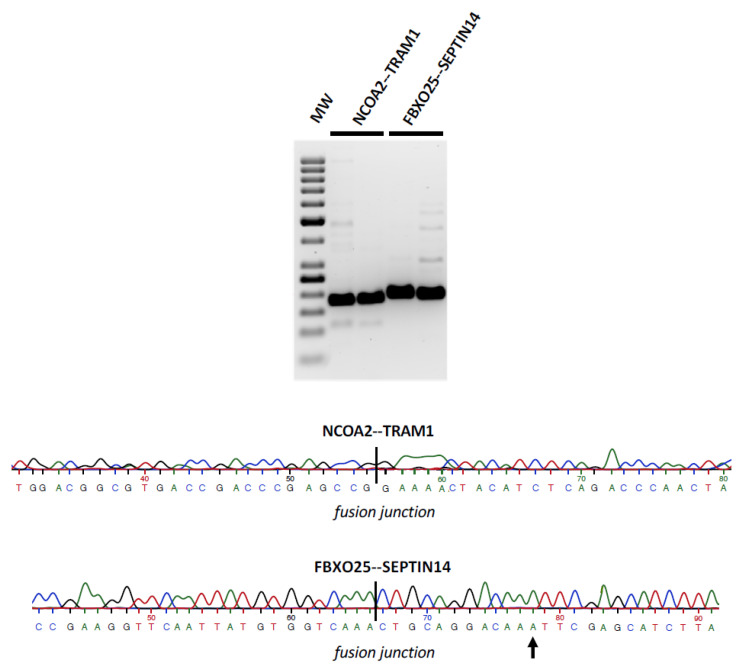
Electrophoresis and sequence of the NC0A2--TRAM1 and FBXO25--SEPTIN14 fusion PCR products. Agarose gel electrophoresis of PCR products generated from cDNA of the cerebellar samples 83 and 86 for the NC0A2--TRAM1 and FBXO25--SEPTIN14 fusions, respectively, using primer pairs specific for each fusion (Section 2 Materials and Methods). MW, molecular weight markers. The electropherograms of the sequenced PCR products, with the fusion junction indicated, is shown. Note the presence of an A nucleotide at position 12 of the SEPTIN14 sequence, indicated by an arrow, instead of the anticipated G nucleotide.

**Figure 4 cancers-14-03134-f004:**
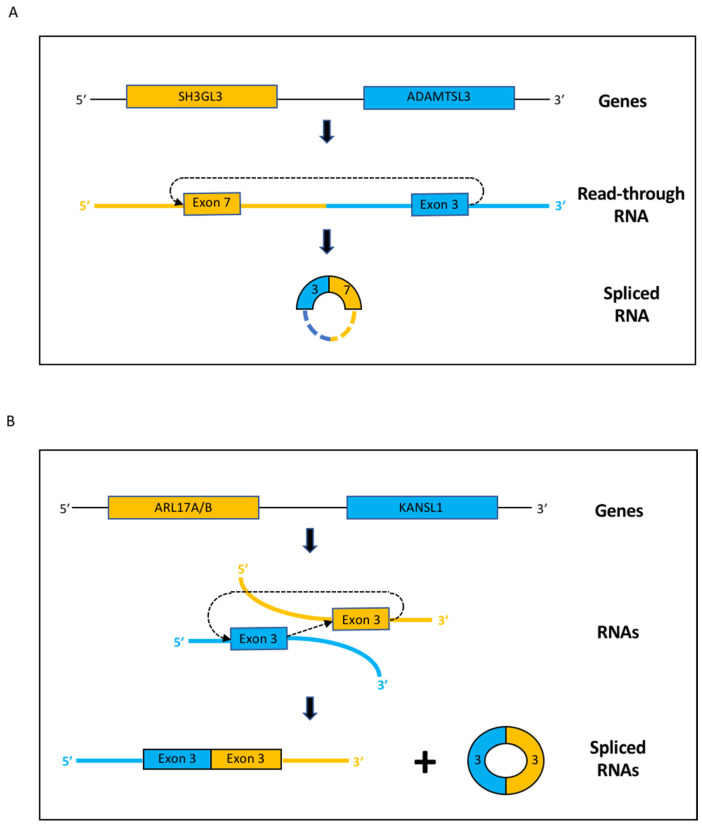
Schematic representation of molecular events leading to the observed ADAMTSL3--SH3GL3 circular fusion and the KANSL--ARL17A/B linear and circular fusions. (**A**) Read-through transcripts from the SH3GL3 gene extend into the downstream ADAMTSL3 gene, followed by back-splicing of exon 3 of ADAMTSL3 to exon 7 of SH3GL3. The resulting circle may or may not include additional ADAMTS3 or SH3GL3 exons, indicated by hatched lines on the diagram. (**B**) Transcripts from the KANSLI1 and ARL17A/B gene bring in proximity the exon 3 of KANSL1 to the exon 3 of ARL17A/B. The 3′ end of KANSLI1 exon 3 trans-splices to the 5′ end of ARL17A/B exon 3 producing the observed KANSLI1--ARL17A/B linear fusion. In cases where a back-splicing event also occurs, with the 3′ end of ARL17A/B exon 3 joining the 5′ end of KANSL1 exon 3, a circular RNA composed of these two exons is produced.

**Table 1 cancers-14-03134-t001:** Selected linear and circular fusions.

Genes in Fusion	Chromosomal Position	Exon number at Fusion Junction	Relative Orientation
*Cerebellum-linear*			
SEPTIN7P14--PSPH	Chr4-Chr7	1(*NR_037630.1*)-4(*NM_004577.4*)	NA
FBXO25--SEPTIN14	Chr8-Chr7	5(*NM_183420.2*)-10(*NM_207366.3*)	NA
NCOA2--TRAM1	Chr8, neighboring genes	1(*NM_006540.4*)-5(*NM_014294.6*)	—> —>
PAUPAR--RCN1	Chr11, neighboring genes	1(*NR_033971.1*)-2(*NM_002901.4*)	—> —>
RPS10--NUDT3	Chr6, neighboring genes	5(*NM_001014.5*)-3(*NM_006703.4*)	—> —>
*Cerebellum-circular*			
ADAMTSL3--SH3GL3	Chr15, neighboring genes	3(*NM_207517.3*)-7(*NM_003027.5*)	—> —>
*Medulloblastoma-linear*			
KANSL1--ARL17A	Chr17, neighboring genes	3(*NM_015443.4*)-3(*NM_001113738.2*)	—> —>
TFG--ADGRG7	Chr3, neighboring genes	3(*NM_006070.6*)-2(*NM_032787.3*)	—> —>
SPSB4--PXYLP1	Chr3, neighboring genes	2(*NM_080862.3*)-2(*NM_001037172.3*)	—> —>
PVT1--CASC8	Chr8, neighboring genes	1(*NR_003367.3*)-6(*NR_117100.1*)	<— —>
*Medulloblastoma-circular*			
ARL17A--KANSL1	Chr17, neighboring genes	3(*NM_001113738.2*)-3(*NM_015443.4*)	—> —>
KANSL1--ARL17A	Chr17, neighboring genes	3(*NM_015443.4*)-3(*NM_001113738.2*)	—> —>
ARL17B--KANSL1	Chr17, neighboring genes	3(*NM_001039083.5*)-3(*NM_015443.4*)	—> —>

NA: not applicable; NCBI accession number provided after exon number; The arrows, —> and —>, depict the relative orientation in the chromosome of the first and second gene found in the fusion.

## Data Availability

The raw RNA-seq data have been submitted to NCBI, with a GEO accession number GSE203174.

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
