# Peer review of "Circular and Fusion RNAs in Medulloblastoma Development"

_cancers, 2022, doi:10.3390/cancers14133134_

Round 1

Reviewer 1 Report

To editors and reviewers
Circular and fusion RNAs in medulloblastoma development
- This is an very interesting manuscript that can be considered for publication in CANCERS. The manuscript is appropriate with aims and scope of journal. This manuscript is complete impressive and innovative which contribute important knowledge for medulloblastoma.
- I suggested some revisions below and after revisions the manuscript can be published.
1) Some citation and references are not precise as MDPI format. Please check and revise.

2) Abstract should be structured format.

3) Introduction need to introduce about the common of medulloblastoma, please refer PMID: 31666838Please also write an imaging paragraph in introduction part to diagnose medulloblastoma, please refer PMID: 32366451.

4) Please add limitation part along with further direction of research related to this topic.

5) Though, "Institutional ethical review was waived for this study", please indicate clearly which institute waived such as your institute or hospital.

Sincerely

Reviewer 2 Report

The authors of the manuscript "Circular and fusion RNAs in medulloblastoma development" have presented interesting data on different types of differentially expressed circular and fusion RNAs between cancerous and normal cerebral tissue samples. The data presented is novel to the research field but needs to clarify some major points before being suitable for publication:

1. The introduction to the manuscript is very short and vague. It needs to elaborate better the significance and relevance of studying circRNAs in medulloblastoma. The different sub-types of medulloblastoma need to be elaborated better, and how studying circRNA in these different sub-types is relevant needs to be better explained.

2.  In line 96-98, the authors mention that for the fusion reads only fraction of total counts per sample was used for comparison. However, the data for read depth in these individual samples is not provided and in the absence of such read normalization quantitative conclusions are weak.

3. In Supp. figure S2A-D, the read counts for the circRNA and the corresponding overlapping mRNA is in the similar range? Is it some mistake in the representation, since the authors claim in line 154-155 that the circular RNA are expressed less than linear polyA mRNAs. A clarification is necessary and corresponding changes in the interpretation of the data.

4. It is important to have some qPCR based validation of the differentially expressed circRNAs in different patient samples.

5. In line 137-140, the authors mention about excluding sample 9 and 10, due to some technical challenges. In that case, the authors should then use the stringent adjusted p-value rather than just the p-value for their analysis, since potential outliers (9,10) were excluded? This will change the interpretation of the data and the conclusions made.

6. The discussion is also rather vague, with no direct interpretation of the data in the light of different sub-types of the cancer tissues? Does the data suggest some differences in circRNA expression between these different sub-types of cancer tissues? Just showing uniquely differentially expressed circRNA like done in figure 1b is not sufficient without proper statistical analysis. This would add another layer of novelty to the manuscript and raise the impact of such precious dataset.

I therefore do not see the manuscript fit for publication in its present state and recommend a major revision with the above mentioned points. Overall, the manuscript has merit however stringent statistics and more diverse comparisons/ interpretations are critical to increase its impact.

Round 2

Reviewer 2 Report

The authors of the manuscript "Circular and fusion RNAs in medulloblastoma development" in their revised version have addressed convincingly all my concerns. I see the manuscript fit for publication in its present form.